# Anthropometric Assessment of General and Central Obesity in Urban Moroccan Women

**DOI:** 10.3390/ijerph19116819

**Published:** 2022-06-02

**Authors:** Natascia Rinaldo, Stefania Toselli, Emanuela Gualdi-Russo, Meriem Khyatti, Amina Gihbid, Luciana Zaccagni

**Affiliations:** 1Department of Neuroscience and Rehabilitation, Faculty of Medicine, Pharmacy and Prevention, University of Ferrara, Corso Ercole I d’Este 32, 44121 Ferrara, Italy; natascia.rinaldo@unife.it (N.R.); luciana.zaccagni@unife.it (L.Z.); 2Center for Exercise Science and Sports, University of Ferrara, 44123 Ferrara, Italy; 3Department of Biomedical and Neuromotor Science, University of Bologna, Via Selmi 3, 40126 Bologna, Italy; 4Institut Pasteur du Maroc, Casablanca 20250, Morocco; meriem.khyatti@pasteur.ma (M.K.); gihbidamina@gmail.com (A.G.)

**Keywords:** health risk, obesity, women, anthropometry, Morocco

## Abstract

In the last few decades, North African countries have faced the nutrition transition, leading to an increase in obesity, exacerbated by an extremely low rate of physical activity (PA). Particular attention must be paid to abdominal obesity (one of the metabolic syndrome criteria), which has been linked to several health problems. The present study aims to investigate the prevalence of overweight/obesity, particularly abdominal obesity, in a sample of urban Moroccan women and to analyze the anthropometric indicators of metabolic syndrome risk among subsamples with different PA and socio-demographic characteristics. Urban Moroccan women living in Casablanca (*n* = 304; mean age 37.4 ± 15.6 years) were recruited for this cross-sectional study. Data concerning socio-demographic variables, PA behavior, and anthropometric measures (height, weight, waist, and hip circumferences) were directly collected. Body mass index, waist-to-hip ratio, waist-to-height ratio, and relative fat mass were computed. Comparisons between women with different socio-demographic characteristics were performed through ANCOVA adjusted for age. The results reveal that 39.4% of the women did not practice any PA. The percentage of women above the cutoffs of risk for general and central obesity was more than half for all the indexes, except for waist-to-hip ratio (WHR), and 19.6% were at a very high risk of health issues. Moreover, being female unmarried, childless, graduates, and students were found to be protective against obesity. In conclusion, Moroccan women have a high level of obesity, especially abdominal, and preventive interventions are needed to reduce the health impact of obesity in this population.

## 1. Introduction

Obesity and overweight are increasing worldwide, leading to increased morbidity and mortality rates [1,2,3]. This increase is marked especially in low-to-middle-income and Middle Eastern and North African (MENA) countries [4,5,6]. In the last few decades, these, in particular, are facing a process called nutrition transition [7,8,9]. This phenomenon is linked to the increasing number of inhabitants, the disparity between social classes, and considerable modification in the diet with large consumption of meat and bread [8,10]. Moreover, reduction in physical activity (PA) and adoption of an inactive lifestyle are also associated with this phenomenon [11]. Data concerning Moroccan residents show a rise in overweight and obesity in children [12] and adults of both sexes, which constitutes a major health problem linked to an increased risk of metabolic syndrome (MetS) [13,14,15,16,17]. However, the categories most at risk are represented by overweight women, especially those living in urban areas [10,18]. According to the 2013–2014 report of the Moroccan High Commission for Planning, the prevalence of overweight and obesity among Moroccan women has increased, respectively, from 29.9% and 16.0% in 2001 to 34.7% and 26.8% in 2011 [19]. Furthermore, Gualdi-Russo et al. [16] reported that 38.7% and 22.6% of Moroccan women were overweight and obese in 2015, respectively. Most previous studies considered only the body mass index (BMI) as an indicator of obesity, while BMI does not always accurately reflect the degree of fatness [20,21]. Accordingly, scientific evidence highlights that the degree of abdominal obesity is more closely tied to health and mortality risks than BMI [22,23,24,25,26]. Therefore, it is fundamental to include in the population studies the measures of abdominal obesity [27] through a series of anthropometric measurements and indices, such as waist circumference (WC), waist-to-hip ratio (WHR), and waist-to-height ratio (WHtR). In this context, the WHO has proposed a combination risk with BMI and WC [23].

Unfortunately, only a few non-recent studies evaluated the prevalence of abdominal obesity in Moroccan women [14,16]. Jafri et al. [14] reported that 67.1% of Moroccan women had a WC of >88 cm and 54% had a WHR of >0.85, 36.2% of the women were overweight and 47.4% were obese. Both central and general obesity were found to increase with age, even if abdominal obesity was reported to be higher already at a younger age [14]. The association between overall and abdominal obesity and sociodemographic factors is not well understood, as it can be a combination of several factors, including but not limited to age, education, lifestyle, cultural beliefs, and economic status [18]. Socioeconomic status is significantly correlated with metabolic syndrome and obesity in Moroccan women [13]. Married women seemed to be more at risk of developing abdominal fat [14,18,28] but the role played by educational level seems controversial [14,28,29]. One of the main factors related to the increase in obesity and fatness appears to be the inadequate PA level and decreased sports practice [28,30]. In particular, Moroccans, especially women, were found to be poorly physically active, and the main determinants for low PA in women seem to be living in an urban environment and being unoccupied [31]. The studies focusing on PA and sports participation in Moroccan women and their relation with overall and abdominal obesity are scarce and contradictory. A study conducted by El Rhazi et al. [18] did not find any significant association between PA and BMI. On the contrary, Rguibi and Belahsen [28] found a significant association between obesity and the time spent in walking activity and traditional sedentary occupations.

Understanding these health-related factors is essential for implementing a health promotion program, especially for women living in urban settings. Thus, the present study aimed to explore the prevalence of overweight/obesity, particularly abdominal obesity, in a sample of urban Moroccan women and to analyze the anthropometric indicators of obesity in subsamples with different PA and socio-demographic characteristics.

## 2. Materials and Methods

### 2.1. Participants

This cross-sectional study was conducted on a convenient sample of 304 Moroccan women living in Casablanca. The inclusion criteria for the enrollment included being female; aged ≥ 18 years; not pregnant; a city resident; and apparently healthy as observed by physical examination. Potential participants were excluded if they had an origin other than Morocco. The study sample was recruited from outpatients with follow-up appointments at public health clinics in Casablanca from May 2017 to October 2020 (one day a month). The rate of refusal to participate in the research (mainly for reasons of time or lack of interest) was around 10%. Upon completing the informed consent process, the number of women, who voluntarily agreed to participate in the research, determined the study sample size. The study had no missing data.

In the beginning, female interviewers completed structured questionnaires with face-to-face interviews to acquire participants’ socio-demographic and behavior data (age, country of birth, civil status, occupational status, number of children, level of education; PA). PA was assessed by computing the 4-level Physical Activity Index (PAI) provided by the General Practice Physical Activity Questionnaire (GPPAQ), which is deemed an easy tool for evaluating PA levels in general practice [32]. Our data analysis also considered PA hours in the last week.

Anthropometric and body composition measurements were directly performed by expert operators using standard techniques and equipment [33,34]. Stature (recorded to the nearest 0.1 cm) and weight (recorded to the nearest 0.1 kg) were measured for participants in light clothing and bare feet; WC was measured at an intermediate level between the lower margin of the lowest rib and the upper margin of the iliac crest, and hip circumference (HC) at the level of the maximum gluteal posterior extension. Both circumferences were measured to the nearest 0.1 cm. More details on the techniques used in anthropometric measurements have been reported previously [35].

We computed the following adiposity indices and body composition: BMI was calculated as weight (in kg) divided by the square of stature (in m). Based on BMI, women were classified as underweight, normal weight, overweight, and obese according to World Health Organization [36] cut-off points. WHR was calculated as WC divided by HC. WHtR was calculated as WC divided by stature. The increased health risk was defined by a cutoff of ≥ 0.85 for WHR and ≥0.50 for WHtR [37,38]. In addition, an increased risk of central obesity has been reported for WC of ≥ 80 cm, and a substantially increased risk for WC of ≥ 88 cm, according to the cutoff proposed by the WHO [24]. The NICE BMI-WC matrix, developed by the National Institute of Health and Clinical Excellence for managing overweight and obesity [39], was applied to assess the obesity risk according to Table 1.

Relative fat mass (RFM) was computed using the equation proposed by Woolcott and Bergman [40]. We used the 35% RFM cutoff for obesity in adult Caucasians [41], corresponding to a BMI of 30 kg/m^2^.

This study represents an extension of a broader European research (EU 7th Framework Programme 2007–2013) conducted on North African resident and immigrant populations [15,16,42,43,44].

All participants gave informed consent prior to enrollment.

Strengthening the Reporting of Observational Studies in Epidemiology (STROBE) Statement [45] was used as a reporting guideline for our observational research. The compiled STROBE checklist was included as a Appendix A.

### 2.2. Statistical Analysis

Descriptive statistics were computed as follows: frequencies and percentages in the case of qualitative variables, means, standard deviations, and 95% confidence interval in the case of quantitative variables. We tested the normality of the variables’ distribution by the Kolmogorov–Smirnov test. The comparisons among the mean values of indicators for the groups of women with different socio-demographic characteristics were performed by ANCOVA, controlling for age. Tukey’s post hoc test was then performed. The significant level was set at *p* < 0.05. Statistical analyses were carried out with STATISTICA software, version 11 (StatSoft, Tulsa, OK).

## 3. Results

The Moroccan sample’s socio-demographic characteristics and PA practice are given in Table 2. The majority of the women were never married, childless, graduated, and practiced PA for less than 1 h a week. The mean number of children was 1.45 ± 2.12 (range 0–10). Almost half of the sample were students. Concerning PA, this sample of women was characterized by very low levels; the PAI indicated that only a quarter should be considered active. In the sample, 10.5% were post-menopausal women (mean menopause age 48.65 ± 4.80 years).

The anthropometric characteristics of the sample are presented in Table 3. The mean value of BMI falls within the overweight range; according to the BMI value, in the sample, 5.3% were underweight, 44.1% were normal weight, 28.3% were overweight, and 22.3% were obese.

The mean values of WC and WHtR were higher than the recommended cutoffs of the criteria of MetS by the WHO [24]. Regarding WC, 38.1% of women had low WC, 16.8% had high WC and almost half of the sample (45.1%) had a very high WC. The mean value of RFM was higher than 35%, the cutoff for obesity in adult Caucasian women [41], corresponding to a BMI of 30 kg/m^2^.

Table 4 shows the percentage of participants at risk of MetS according to the anthropometric indicators considered. The indicator with the lowest percentage was WHR (40.1%), and the indicator with the highest percentage was WC (62.9%). More than half of the sample was affected by central obesity by WC and WHtR, while considering the WHR index, the prevalence rate of central obesity was less than half. Of the 150 participants classified as normal weight using BMI, the prevalence of central obesity was 30.4, 24.3, and 18.9% based on WC, WHtR, and WHR, respectively.

Only 12.9% of overweight women had a low-risk WC, while the majority of the obese participants had a very high-risk WC. Of the 118 women classified as normal fat using RFM < 35%, the prevalence of central obesity was 7.2% based on both WC and WHR, while no woman in this subsample showed an increased risk according to WHtR.

Using the NICE BMI-WC composite index, 39.5% of the women in the sample had a high or very high health risk, while nearly half of the sample had no increased health risk (bottom of Table 4). As only one woman was very obese based on BMI, the participants in the category very high risk had a very high WC.

Table 5 shows the results of the analysis of covariance. After adjusting for age, never married, childless, with university degrees, and student women had significantly lower mean values of BMI, WC, WHtR, and RFM than other groups of women. The only indicator that did not reach the level of statistical significance was the WHR. Tukey’s post-hoc test highlighted the differences between women with a degree and the other groups for educational level and between unemployed and the other groups for occupational status.

No significant effect was detected according to the PA practiced in the last week.

## 4. Discussion

This study examined the overall and central obesity in a sample of urban Moroccan women and the association of anthropometric indicators of MetS risk with socio-demographic and PA characteristics. The sample examined consisted of women who were residents in an urban area and precisely in Casablanca, the largest city in Morocco. Although the subsamples were numerically quite balanced, most of these women were unmarried and childless. They were also predominantly university graduates. Only about a third of them were unemployed.

All statistical comparisons between the subsamples in this study were conducted by adjusting for age, given its influence on body composition with an increase in body fat mass with aging [46]. The significantly higher values of overall and abdominal obesity (except for WHR) in married women of our sample compared to unmarried ones are well reflected in cultural reasons, as it is believed that female fatness is attractive in North African populations [14]. However, although an ancestral image of beauty, a curvy body, is still found to be associated with low socioeconomic status or isolation in North African women, the desire to be thinner is now emerging [47]. Another factor that significantly affected the overall and abdominal obesity (except for WHR) of the study sample seems to be parity, with significantly higher general and abdominal obesity levels in women with kids. Although the influence of parity on obesity is still controversial in the emerging literature [27,48], Bladeau et al. [49] attested to an increase in abdominal adipose tissue with increasing parity. The prevalence of overall and central obesity was significantly lower (except for WHR) in female graduates than in women with lower levels of education, according to previous studies [14,18]. The association between adiposity indexes and occupational status was weak. Only BMI was significantly lower in female students, the only ones to show average index values well below the risk cutoffs.

Other lifestyle factors, such as particular sedentary behavior, which in the sample examined exceeds 58% (wholly inactive/moderately inactive women), can be responsible for the observed overall and abdominal obesity consistent with studies from the literature documenting the effects of PA in individuals of all ages [50,51,52]. In the present study, however, the generalized situation of inactivity or reduced PA (small size of active subsample) did not highlight the effects of PA on the obesity indicators.

The picture we drew for the sample of Moroccan women studied is consistent with the worldwide trend in the obesity epidemic with total obesity that has affected all populations of the world and different age groups [1,2,53,54], with more than 50 obesity-related comorbidities [55]. A major cause of the obesity epidemic is, in addition to diet, physical inactivity or reduced PA [56,57]. In particular, a recent study of BMI trends in populations worldwide found that in a nutrition transition, the increase in nutrient-poor, energy-dense foods can result in an increase in BMI and worse health outcomes across the lifespan [54], as in the studied population. Adiposity indices are considered predictors of health risk. The literature shows that increased health care use is associated with all measures of adiposity, although this association is attenuated in older adults (aged 75+) [58]. More generally, morbidity and mortality increase with increasing obesity [59,60].

The most widely used method for assessing obesity is BMI, considering at risk all those who have an index greater than 25 kg/m^2^ according to the WHO guidelines. However, BMI can only be a proxy for the level of adiposity, so much so that an anthropometric assessment of abdominal or central obesity is deemed to be a better predictor of all-cause mortality than BMI and that combined use of various central adiposity indices is likely to be even better [61]. Using BMI alone for measuring obesity can be misleading because it does not distinguish between fat and lean body mass, poorly depicting the body fat percentage of an individual [62]. Moreover, BMI does not give any information about the location of body fat. The accumulation of fat in the abdominal area can be assessed by other indicators, such as WC, WHR, WHtR, which we applied in our study. In short, the literature has highlighted the importance of the localization of body fat, in addition to the total amount of body fat [63]. The accumulation of fat in the abdominal region involves an increased risk of health issues. In particular, this accumulation is associated with the risk of coronary heart disease and hypertension, insulin resistance, and diabetes [38,64,65]. Centrally obese individuals with normal BMIs but atherogenic dyslipidemia have a similar, and perhaps higher, risk of mortality than those who are centrally obese and overweight or obese based on their BMI [61].

The prevalence of obesity in the sample of Moroccan women examined varied depending on the indicator of obesity used, confirming the concept that obesity is a complex disease that cannot be evaluated by a single tool [58]. Half of the sample tested was found to be in the overweight/obese category according to BMI, and an even higher proportion of women at MetS risk were found by the analysis of all other indicators of central obesity (WC, WHtR, and BMI-WC matrix, apart from WHR). Moreover, about 60% of women had RFM above the cutoff. The health risk analysis of the group with “normal” BMIs showed the presence of women who exceeded the cutoffs according to WC, WHR, and WHtR, and vice versa, some women from the overweight group were found to be below the cutoffs. We found more than half of the sample were at an increased risk according to WHtR and BMI-WC matrix.

An important result is that, in line with the findings in the MENA region [27,66,67], we found a higher prevalence of abdominal obesity according to all indicators (except WHR) than overall obesity by BMI in this sample. This pattern probably resulted from the fact that more than 30% of normal-weight women had abdominal obesity according to WC (more than 20% according to WHtR). This is not surprising, and it is another proof that BMI is just an approximate indicator of the degree of adiposity [61], and individuals with a normal BMI may have a percentage of body fat higher than 30% [68]. More than half of the normal-weight women have a body fat percentage higher than 30% in our sample. This aspect deserves great consideration because, when body fat accumulates at the abdominal level, it increases health risk by positively correlating with metabolic abnormalities [69]. Although BMI may be useful for population screening, considering only the BMI for measuring obesity is potentially insufficient to identify those at increased risk for associated conditions, particularly cardio-metabolic disease, because there are metabolically healthy obese persons and metabolically unhealthy persons with normal weight [69].

In a recent study [69], WHtR was the best predictor of DXA-derived whole-body fat percentage and visceral adipose tissue mass in females and males. However, WC can be an alternative predictor, unlike WHR. Consistent with another study [70], our findings show that the prevalence of individuals at risk through WHtR is similar to that obtained with the combined BMI-WC matrix (57.0% for WHtR vs. 53.1% for BMI-WC matrix integrating individuals at increased/high/very high risk). Which of the adiposity indexes is to be preferred is still a matter of discussion in the literature [71]; although, there seems to be a general preference for WHtR because of its independence from BMI, the use of the same cutoff (0.5) regardless of sex and population, the need for minimum equipment and the possibility of being monitored directly by the patient [61], with the simple suggestion that the WC should not exceed half the stature. In essence, although the results of the present study cannot state which of the obesity indicators is the best, they can confirm, on the one hand, the inadequacy of BMI alone and, on the other hand, the unsatisfactory WHR response in comparison with the other central obesity indicators.

The prevalence of obesity in our study confirms the previous report of the Moroccan Ministry of Health, indicating that 22% of women aged ≥20 years have a BMI of ≥30 kg/m^2^ [72]. Although the incidence of overweight/obesity in Morocco is in line with the increasing trend in the mean BMI from Southern to Northern Africa [17], the prevalence of obesity in Morocco is lower than in other MENA countries. Studies on Tunisian women found an overall obesity prevalence variable from 50.2% [73] to 80.6% [47]. In Egypt, the incidence of women with a BMI of ≥ 30 kg/m^2^ was 49.5% according to a national survey conducted in 2019 [74]. Central obesity (WC ≥ 80 cm) prevalence was even higher than overall obesity in our sample (62.9%) but lower than that found in Tunisian women (80.6% according to [27]). In Morocco, a lower prevalence of total obesity was found in the rural area [75,76] than in urban areas ([14,16] and present study), consistent with the literature findings showing that excess weight is more prevalent in women living in urban versus rural areas [29]. However, in a previous study in Casablanca, Jafri et al. [14] reported a much higher incidence of overweight/obesity (83.6%) than we found in the present study (50.7%), likely due to socioeconomic differences between the two study samples, since the first study referred exclusively to women from the neighborhoods of Casablanca. There is no doubt that several lifestyle-related factors may be involved in the different obesity rates and even more so in the case of comparisons between the urban and rural environment of residence, although ethnicity and genetic factors could drive regional differences [27,77].

Our findings have some policy implications, highlighting the need to monitor and reduce female obesity to have health benefits. We emphasize following the WHO [78] recommendations to reduce obesity and the latest WHO guidelines [79] regarding the practice of PA, and evaluating at the national level the interventions to be carried out, according to a normative and cultural perspective. Further research should be conducted to better estimate obesity in the Moroccan population (including the male segment) and related comorbidities.

Limitations of this study include its cross-sectional design and the use of a convenience sample of just females. Future research may test whether the patterns found in this study are confirmed in random, larger samples of the Moroccan population. Another limitation of the study is not measuring dietary intakes of total energy and other nutrients. Despite these limitations, the study highlighted some findings of interest in the Moroccan female population, allowing the collection of new information on abdominal obesity.

The main strength of our study was the use of anthropometric variables directly measured on participants according to standardized techniques. Importantly, our data do not refer to a specific age group but a wide range of ages, thus allowing a broader vision of the problem.

## 5. Conclusions

With reference to the aims of this study, obesity was found to be a widespread condition in the sample examined, with more than half of the Moroccan women examined having indices of overall and central obesity beyond the cutoffs and being at risk of health issues. Due to the presence of normal-weight women with central obesity, the prevalence of central obesity was much higher than general obesity.

Furthermore, the comparison of anthropometric indicators of obesity among the subsamples indicated that unmarried, childless, graduates, or student women were less likely to be obese. Although the causes of the detected obesity have not been fully elucidated, the inactivity or low PA level of the study participants should be strongly underlined.

In light of the findings of the present investigation, it is clear that indicators of abdominal adiposity, in addition to BMI, should be included in population health surveillance. Adiposity indices and body composition parameters can provide important prognostic indications of a person’s health. In particular, WHtR and WC appear to be effective means of controlling the obesity epidemic.

Obesity, especially central obesity, is a major cause of poor health, being a risk factor for many serious diseases. Therefore, more research is needed to assess the different actions and policies capable of reducing obesity in Morocco, thereby reducing the health impact of obesity.

## Figures and Tables

**Table 1 ijerph-19-06819-t001:** Obesity risk groups using the NICE BMI-WC matrix [39].

	BMI: Normal	BMI: Overweight	BMI: Obese	BMI: Very Obese
**Low WC**	No increased risk	No increased risk	Increased risk	Very high risk
**High WC**	No increased risk	Increased risk	High risk	Very high risk
**Very high WC**	Increased risk	High risk	Very high risk	Very high risk

**Table 2 ijerph-19-06819-t002:** Socio-demographic characteristics and PA practice of the Moroccan sample (*n* = 304).

Variable	%
**Marital status**	
never married	52.3
married	47.7
**Number of children**	
childless	56.3
1 or more	43.5
**Educational level**	
no schooling	14.9
primary school	10.6
middle school	7.3
high school	9.6
university	57.6
**Occupational status**	
unemployed	28.0
employed	29.2
student	42.8
**Physical activity during last week**	
PA < 1 h	75.5
1 ≤ PA < 3	8.4
PA ≥ 3	16.1
**PAI level**	
inactive	39.4
moderately inactive	19.0
moderately active	16.3
active	25.3

**Table 3 ijerph-19-06819-t003:** Anthropometric characteristics of the Moroccan sample (*n* = 304).

Variables	Mean	SD	95% C.I.
Age (yrs)	37.4	15.6	35.7–39.2
Stature (cm)	162.0	7.2	161.2–162.8
Weight (kg)	67.4	14.1	65.8–69.0
WC (cm)	86.9	16.9	84.9–88.9
HC (cm)	105.1	12.9	103.6–106.6
**Indices**			
BMI (kg/m^2^)	25.74	5.34	25.14–26.35
WHR	0.82	0.10	0.81–0.84
WHtR	0.54	0.11	0.52–0.55
RFM (%)	37.26	7.83	36.35–38.17

**Table 4 ijerph-19-06819-t004:** Percentage of women at risk of MetS according to anthropometric indicators in Moroccan women (*n* = 304).

Indicators	%
BMI ≥ 25 kg/m^2^	50.7
WC ≥ 80 cm	62.9
WHR ≥ 0.85	40.1
WHtR ≥ 0.50	57.0
RFM ≥ 35%	61.2
**BMI-WC health risk**	**%**
No increased risk	46.8
Increased health risk	13.6
High risk	19.9
Very high risk	19.6

**Table 5 ijerph-19-06819-t005:** Anthropometric indicators of metabolic syndrome risk by socio-demographic and PA variables (*n* = 304).

	BMI (kg/m^2^)		WC (cm)		WHR		WHtR		RFM (%)	
**Marital status**	mean	SD	mean	SD	mean	SD	mean	SD	mean	SD
never married	23.34	4.59	77.7	13.3	0.79	0.09	0.48	0.08	32.83	7.02
married	28.40	4.95	96.6	15.0	0.86	0.09	0.60	0.10	41.83	5.73
*F (ANCOVA)*	*19.908*		*16.439*		*1.243*		*16.709*		*20.843*	
*p value*	** *<0.001* **		** *<0.001* **		*0.266*		** *<0.001* **		** *<0.001* **	
**Number of children**	mean	SD	mean	SD	mean	SD	mean	SD	mean	SD
childless	23.52	4.68	78.2	13.3	0.79	0.09	0.48	0.08	33.17	7.01
1 or more	28.77	4.79	98.1	14.5	0.87	0.09	0.61	0.09	42.46	5.38
*F (ANCOVA)*	*19.906*		*19.051*		*2.503*		*19.523*		*21.695*	
*p value*	** *<0.001* **		** *<0.001* **		*0.115*		** *<0.001* **		** *<0.001* **	
**Education**	mean	SD	mean	SD	mean	SD	mean	SD	mean	SD
no schooling	27.13	5.73	100.4	15.3	0.89	0.08	0.62	0.10	42.92	5.72
primary school	27.97	5.85	97.1	15.2	0.88	0.08	0.61	0.10	41.96	6.54
middle school	29.82	5.04	97.7	13.0	0.85	0.08	0.62	0.10	42.76	5.15
high school	28.84	3.73	94.5	13.0	0.86	0.09	0.59	0.08	41.29	4.62
university	24.00	4.77	78.9	14.1	0.79	0.09	0.48	0.09	33.48	7.18
*F (ANCOVA)*	*4.571*		*3.959*		*1.684*		*4.309*		*4.454*	
*p Value*	** *0.001* **		** *0.004* **		*0.154*		** *0.002* **		** *0.001* **	
**Occupational status**	mean	SD	mean	SD	mean	SD	mean	SD	mean	SD
unemployed	28.53	4.80	94.3	16.5	0.85	0.11	0.59	0.11	40.82	6.67
employed	26.06	4.96	86.9	16.3	0.83	0.09	0.53	0.10	37.23	7.34
student	22.85	4.40	75.6	11.8	0.78	0.09	0.46	0.07	31.83	6.73
*F (ANCOVA)*	*4.447*		*1.066*		*0.085*		*1.433*		*1.274*	
*p value*	** *0.013* **		*0.346*		*0.918*		*0.241*		*0.282*	
**PA last week**	mean	SD	mean	SD	mean	SD	mean	SD	mean	SD
PA < 1 h	25.90	5.62	87.3	17.9	0.83	0.10	0.54	0.11	37.23	8.21
1 ≤ PA < 3 h	25.55	5.17	84.8	17.2	0.81	0.10	0.53	0.12	36.62	7.66
PA ≥ 3 h	25.10	4.67	85.4	13.9	0.81	0.08	0.53	0.10	37.03	6.90
*F (ANCOVA)*	*0.051*		*0.094*		*0.257*		*0.018*		*0.419*	
*p value*	*0.950*		*0.910*		*0.774*		*0.982*		*0.740*	

## Data Availability

Authors will provide data to all interested parties upon reasonable request.

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
