# Peer review of "Anthropometric Assessment of General and Central Obesity in Urban Moroccan Women"

_ijerph, 2022, doi:10.3390/ijerph19116819_

Round 1

Reviewer 1 Report

The study is well designed and the results and discussions are appropriate to the study design.

I send small review suggestions:

In the methods, it is suggested to define the criteria to define "being apparently healthy" (line 94).

Has any test been performed to verify that the distribution of the data approximates a normal distribution? If positive, inform which test was performed. If no normality test has been performed, I suggest that it be done to make sure the "average" is the best measure of central location. If the distribution is not close to normal, we suggest using the "median" as the central measure.

Author Response

The study is well designed and the results and discussions are appropriate to the study design.

I send small review suggestions:

Answer: we thank you for the comprehensive and positive review of our manuscript.

In the methods, it is suggested to define the criteria to define "being apparently healthy" (line 94).

Answer: we added "as observed by physical examination" to define the criteria used.

Has any test been performed to verify that the distribution of the data approximates a normal distribution? If positive, inform which test was performed. If no normality test has been performed, I suggest that it be done to make sure the "average" is the best measure of central location. If the distribution is not close to normal, we suggest using the "median" as the central measure.

Answer: Thank you for your remarks. We used the Kolmogorov-Smirnov test to determine whether the sample distribution had the characteristics of a normal distribution (line 154). In this case, the distribution was found to be normal.

Reviewer 2 Report

Thanks for your valuable work. Comments have been included to improve the article. I hope the article will be improved by applying the comments

Introduction

Authors mention that  "analyzing the relationships of nutritional status with PA and socio-demographic factors" is a second aim of the current study but there was no mention for nutritional status in the method or in the result.

Method

Did you ask whether women were postmenopausal or not, do you have percentage of postmenopausal women in your participants? Please if you have it included it

Result

In line 159, 164, 168, 175 in the result and in table 4 title; it is preferred to change word subject with women or participants

Discussion

  • In line 201 an abbreviation was given (SES) what is refer to?

  • "Significantly lower mean values of overall and central obesity resulted in female graduates compared to women with lower levels of education." There is no literature support this finding; no published paper linked between education levels and obesity risk

  • In line 223 the authors mentioned that "the studied population had an unhealthy nutritional transition" based on what authors maid this assumption as there is no refer to dietary intake of the study population

Author Response

Thanks for your valuable work. Comments have been included to improve the article. I hope the article will be improved by applying the comments

Answer: Thank you for your constructive comments that helped us to improve the manuscript.

Introduction

Authors mention that  "analyzing the relationships of nutritional status with PA and socio-demographic factors" is a second aim of the current study but there was no mention for nutritional status in the method or in the result.

Answer: The second aim of the study was changed according to the comment of reviewer 4 to:  “to analyze anthropometric indicators of obesity in subsamples with different PA and socio-demographic characteristics” (lines 95-97). The expression "nutritional status” has been deleted.

Method

Did you ask whether women were postmenopausal or not, do you have percentage of postmenopausal women in your participants? Please if you have it included it

Answer: We have included the requested information on lines 167-168.

Result

In line 159, 164, 168, 175 in the result and in table 4 title; it is preferred to change word subject with women or participants

Answer: We have changed the word “subjects”, as you suggested.

Discussion

  • In line 201 an abbreviation was given (SES) what is refer to?

Answer: The abbreviation SES has been replaced with the extended-term “socioeconomic status” (line 225).

  • "Significantly lower mean values of overall and central obesity resulted in female graduates compared to women with lower levels of education." There is no literature support this finding; no published paper linked between education levels and obesity risk.

Answer: In this case, we are reporting the results of our study. However, we have changed the phrase by adding some references from the literature, as you suggested (line 234).

  • In line 223 the authors mentioned that "the studied population had an unhealthy nutritional transition" based on what authors maid this assumption as there is no refer to dietary intake of the study population

Answer: We have now changed the sentence to make the meaning clearer (lines 249-253). Thank you.

Reviewer 3 Report

This manuscript aims to explore the prevalence of overweight/obesity, particularly abdominal obesity, in a sample of urban Moroccan women. It also aims to analyze the relationships of nutritional status with physical activity and socio-demographic factors. 

Major Concerns

  • Nutritional status is never assessed in this study. Nutritional status would refer to measures of dietary intake and diet quality. I believe the authors actually mean weight status or body composition status instead of nutritional status. The term "nutrition" shouldn't even be used outside of describing the nutrition transition.
  • The Purpose of the study is never detailed in the Abstract.
  • The font size changes in lines 56-60.
  • The Statistical Analysis section could be written in one paragraph. It's oddly divided into four paragraphs, three of which are single sentences.
  • In the Results, Metabolic Syndrome (MetS) is all of a sudden introduced. It is also utilized in the first sentence of the Discussion as if it were a part of the aim of the study. This is not true based on what was written in the Introduction. MetS should be discussed in the Introduction if it's going to be analyzed in the Results section.
  • The English language is not utilized correctly in many instances. For example, lines 207-208. "Significantly lower mean values of overall and central obesity (except for WHR) resulted in female graduates compared to women with lower levels of education." 

Minor Concerns

  • Line 145. The mean value of children? The mean number of children or offspring would be better terminology to use.
  • Capitalization of the various Variables is inconsistent in both Table 2 and Table 5.
  • Some of the variables in Table 3 utilize two decimal points for values while others use one decimal point. This is inconsistent. Please stay consistent throughout all tables including Table 5.

Overall, significant revisions are needed for this manuscript before resubmission for another review for publication.

Author Response

This manuscript aims to explore the prevalence of overweight/obesity, particularly abdominal obesity, in a sample of urban Moroccan women. It also aims to analyze the relationships of nutritional status with physical activity and socio-demographic factors. 

Answer: Thank you for reviewing our manuscript and for your comments to improve it.

Major Concerns

  • Nutritional status is never assessed in this study. Nutritional status would refer to measures of dietary intake and diet quality. I believe the authors actually mean weight status or body composition status instead of nutritional status. The term "nutrition" shouldn't even be used outside of describing the nutrition transition.

Answer: Nutritional status can be assessed by various methodologies (Gibson, 2005; Huhmann, 2011) and anthropometric methodologies are widely used in epidemiological research (see, e.g. the WHO assessment of malnutrition, 2017) but we have replaced this expression with terms that do not give rise to misinterpretation, following your guidance (lines 95-97).

  • The Purpose of the study is never detailed in the Abstract.

Answer: Thanks for your notice. Now we have added it. Lines 27-30. 

  • The font size changes in lines 56-60.

Answer: We have changed the font size.

  • The Statistical Analysis section could be written in one paragraph. It's oddly divided into four paragraphs, three of which are single sentences.

Answer: done. Now the Statistical Analysis section is written in one paragraph.

  • In the Results, Metabolic Syndrome (MetS) is all of a sudden introduced. It is also utilized in the first sentence of the Discussion as if it were a part of the aim of the study. This is not true based on what was written in the Introduction. MetS should be discussed in the Introduction if it's going to be analyzed in the Results section.

Answer: We have now introduced the Metabolic Syndrome in the abstract (line 29) and the Introduction (lines 54-55).

  • The English language is not utilized correctly in many instances. For example, lines 207-208. "Significantly lower mean values of overall and central obesity (except for WHR) resulted in female graduates compared to women with lower levels of education." 

Answer: The entire text has been rechecked for English including the indicated sentence.

Minor Concerns

  • Line 145. The mean value of children? The mean number of children or offspring would be better terminology to use.

Answer:  We have changed the “mean value of children” to “mean number of children” (line 164-165).

  • Capitalization of the various Variables is inconsistent in both Table 2 and Table 5.

Answer: Now the capitalization of the variables in tables 2 and 5 is consistent.

  • Some of the variables in Table 3 utilize two decimal points for values while others use one decimal point. This is inconsistent. Please stay consistent throughout all tables including Table 5.

Answer: Thank you for allowing us to explain this point. We have reported a different types of variables in the tables: anthropometric measurements and anthropometric indices. While for the former we cannot report more decimal places than those measured, for the latter obtained by calculation and reported in absolute value or with different units (eg kg/m2 for BMI) we referred to classifications found in the literature with two decimal places. However, we have now provided the unit of measurement of anthropometric variables in the methods (lines 119-120 and 123-124) and divided the table to separate indices from other variables. Now table 5 is consistent with table 3.

Overall, significant revisions are needed for this manuscript before resubmission for another review for publication.

References

Gibson, R.S. (2005) Principles of Nutritional Assessment. 2nd Edition, Oxford University Press Inc., New York.

Huhmann M.B. (2011) Nutrition Status. In: Schwab M. (eds) Encyclopedia of Cancer. Springer, Berlin, Heidelberg. https://doi.org/10.1007/978-3-642-16483-5_4179.

WHO. Guideline: assessing and managing children at primary health-care facilities to prevent overweight and obesity in the context of the double burden of malnutrition. Updates for the Integrated Management of Childhood Illness (IMCI). Geneva: World Health Organization; 2017. Licence: CC BY-NC-SA 3.0 IGO

Reviewer 4 Report

Dear Authors, I am so interested in an opportunity to review a topic such as „Adiposity and health risk in urban Moroccan women“.

Considering that the authors carried out a single cross-sectional study, I found the manuscript interesting. However, I  have a revision request. I suggest the authors to use STROBE checklist in reporting their cross-sectional study. Please, add the STROBE checklist as a supplementary material and cite it through the main text. You can download the checklist from this link https://www.strobe-statement.org/

Major changes

Therefore, I have comments concerning the design of the study, the organisation of the study, and the description of the Methodology. These major concerns in the manuscript can be answered in accordance with STROBE checklist.

Minor changes

I would recommend changing the title of the paper as the Authors did not measure „Adiposity“ and „health risk“. In my opinion, the terms „Body Weight Status“, „Anthropometric Profiles“, etc. would be more appropriate.

The Authors pointed out that they „analyzed the relationships of nutritional status with PA and sociodemographic factors“. It remains unclear how the Authors assessed the relationships? What studies have been carried out to determine the nutritional status? I would recommend substantial adjustments to the aims of the study. I would also suggest changing “sociodemographic factors” to “sociodemographic characteristics.”

L 145: Authors wrote: “The mean value of children was 1.45±2.12”. I would suggest adjusting the data as 1.45 part of the child can't be present (do authors write about people?).

Table 2: Can „Employment Status“ be a “student”?

Table 3: It seems there is an appropriation to move away values such as „minimum“ and „maximum“ and submit 95% CI (Confidence Intervals).

The Conclusions must be adjusted and written in such a way as well as main findings fully reflect the aims of this study.

Kind Regards

Author Response

Dear Authors, I am so interested in an opportunity to review a topic such as „Adiposity and health risk in urban Moroccan women“.

Considering that the authors carried out a single cross-sectional study, I found the manuscript interesting. However, I  have a revision request. I suggest the authors to use the STROBE checklist in reporting their cross-sectional study. Please, add the STROBE checklist as supplementary material and cite it through the main text. You can download the checklist from this link https://www.strobe-statement.org/

Answers: Thank you for reviewing our manuscript and for your positive assessment of our study. As you suggested, we have added the STROBE checklist as supplementary material, citing it in the text (lines 147-149). 

Major changes

Therefore, I have comments concerning the design of the study, the organisation of the study, and the description of the Methodology. These major concerns in the manuscript can be answered following the STROBE checklist.

Answers: we answered your major concerns following the STROBE checklist, as you suggested.

Minor changes

I would recommend changing the title of the paper as the Authors did not measure „Adiposity“ and „health risk“. In my opinion, the terms „Body Weight Status“, „Anthropometric Profiles“, etc. would be more appropriate.

Answers: As suggested, we changed the title to “Anthropometric assessment of general and central obesity in urban Moroccan women”

The Authors pointed out that they „analyzed the relationships of nutritional status with PA and sociodemographic factors“. It remains unclear how the Authors assessed the relationships? What studies have been carried out to determine the nutritional status? I would recommend substantial adjustments to the aims of the study. I would also suggest changing “sociodemographic factors” to “sociodemographic characteristics.”

Answers: Thank you for the remark. We changed “sociodemographic factors” to “sociodemographic characteristics” and adjusted the second aim of the study to:  “to analyze the anthropometric indicators of obesity in subsamples with different PA and socio-demographic characteristics” (lines 95-97). The definition of nutritional status depends on the discipline (Huhmann, 2011) and anthropometric measurements are often used as an assessment of nutritional status in epidemiological studies due to the ease of field assessment, noninvasiveness, inexpensive instrumentation (Thornton and Villamor, 2016; Bhattacharya et al, 2019; WHO, 2017). However, taking into account your criticism, we have replaced the expression "nutritional status" with others (“anthropometric indicators of obesity”) that do not give rise to misunderstanding (lines 95-96).

L 145: Authors wrote: “The mean value of children was 1.45±2.12”. I would suggest adjusting the data as 1.45 part of the child can't be present (do authors write about people?).

Answers: changed to “the mean number of children was 1.45±2.12” (lines 164-165). Thank you.

Table 2: Can „Employment Status“ be a “student”?

Answers: You are right. We think it is more correct “occupational status”. We changed the term throughout the manuscript.

Table 3: It seems there is an appropriation to move away values such as „minimum“ and „maximum“ and submit 95% CI (Confidence Intervals).

Answers: Confidence intervals were entered, as suggested.

The Conclusions must be adjusted and written in such a way as well as main findings fully reflect the aims of this study.

Answers: as you suggested, we have revised the conclusions (lines 356-361) to better reflect the aims of the study. Thank you.

References

Bhattacharya, A.; Pal, B.; Mukherjee, S.; et al. Assessment of nutritional status using anthropometric variables by multivariate analysis. BMC Public Health 2019, 19, 1045 . https://doi.org/10.1186/s12889-019-7372-2

Huhmann M.B. (2011) Nutrition Status. In: Schwab M. (eds) Encyclopedia of Cancer. Springer, Berlin, Heidelberg. https://doi.org/10.1007/978-3-642-16483-5_4179.

Thornton K, Villamor E. Nutritional Epidemiology. Encyclopedia of Food and Health. 2016 Jan 1;104–7.

WHO. Guideline: assessing and managing children at primary health-care facilities to prevent overweight and obesity in the context of the double burden of malnutrition. Updates for the Integrated Management of Childhood Illness (IMCI). Geneva: World Health Organization; 2017. Licence: CC BY-NC-SA 3.0 IGO

Round 2

Reviewer 3 Report

Thank you for making the edits to your manuscript in response to my comments as well as the other reviewer.

Reviewer 4 Report

Dear Authors,   You answered all my comments. Personally, I was interested in reading this manuscript. I have a recommendation to accept the paper.   Best Regards